# Exploring Readiness towards Effective Implementation of Safety and Health Measures for COVID-19 Prevention in Nakhon-Si-Thammarat Community-Based Tourism of Southern Thailand

**DOI:** 10.3390/ijerph191610049

**Published:** 2022-08-15

**Authors:** Apirak Bumyut, Sasithorn Thanapop, Dusanee Suwankhong

**Affiliations:** 1Department of Environmental Health and Technology, School of Public Health, Walailak University, Thasala 80160, Na Khon Si Thammarat, Thailand; 2Department of Community Public Health, School of Public Health, Walailak University, Thasala 80160, Na Khon Si Thammarat, Thailand; 3Department of Public Health, Faculty of Health and Sports Science, Thaksin University, Pa Phayom 93210, Phatthalung, Thailand

**Keywords:** readiness, safety, health, COVID-19 prevention, community-based tourism

## Abstract

Thailand’s community-based tourism (CBT) faces a challenging adaptation in response to COVID-19 prevention. This study aimed to assess the readiness for effective implementation of the Safety and Health Administration (SHA) for COVID-19 prevention in the tourism community. A qualitative approach was adopted for this study. Three communities covering all types of CBT in Nakhon-Si-Thammarat province, southern Thailand were purposively chosen. Fifteen key informants were invited to participate in the study. Semi-structured in-depth interviews were conducted, and the data were analysed using the thematic analysis method. The readiness stage was assigned by consensual comprehensive scores. The overall readiness of CBT is pre-planning stage, a clear recognition of the SHA benefit, and there are sufficient resources for implementation. At this stage, there is no planning because the business owners feel that they have inadequate knowledge about the SHA protocol. Another main barrier is having limited accessibility for SHA information which mainly provides through with technology platform. The CBT owner needs to improve public health-based knowledge, technology and cooperation skills to operate SHA efficiently. However, in order to embed SHA to the CBT, tourism and public health organisation should provide suitable methods at the initiation stage by considering the community readiness and need.

## 1. Introduction

The World Health Organization (WHO) announced COVID-19 as a Public Health Emergency of International Concern (PHEIC) on 31 January 2020, and a pandemic on 11 March 2020 [1]. The COVID-19 pandemic continues to affect many economic sectors worldwide, especially in developing countries [2]. It has even led to economic crises in the tourism and hospitality sectors [3,4,5]. The current global pandemic has significantly affected the tourism sector’s capacity to function translating into the loss of jobs and reduced incomes for tourism and hospitality businesses [2,6]. Likewise, the COVID-19 outbreak in Thailand has affected the country’s tourism and service sectors, which are the main sources of country income [7]. Therefore, tourism entrepreneurs should be able to adjust their service models to take into account the health and safety of their customers amid the pandemic [8,9].

According to the World Travel and Tourism Council (2020), the crisis is an opportunity to reconsider the growth of the future tourism industry [10]. The government needs to deliberate the lasting implications of such a crisis by supporting and promoting public health management and aiming for a sustainable and resilient tourism economy. The Tourism Authority of Thailand (TAT) adopted the Safety and Health Administration (SHA) standards, the COVID-19 prevention strategies for public health, and social measures based on personal hygiene, workplace sanitation, and health risk communication, for various types of service in the tourism destination [11,12,13]. The SHA standard is a cooperative project of the Ministry of Public Health, and a private organisation in partnership with the government in the tourism industry. The SHA objectives are to ensure the health standard and quality of services for tourists are saved and satisfied with the tourism services based on hygiene practices and environmental health programmes.

CBT in Thailand is the tourism of local residents that takes into account the sustainability of the environment, society, and culture. The direction is set up and managed by the community for the community. Thus, the community has the ownership role to manage and care for visitors [14]. The analysis of income distribution through CBT when compared to the induced impact of CBT on the total tourism of the country found that the induced impact of CBT between 2018 and 2019 was higher than the country’s overall tourism business. CBT can have a higher impact on the income of the country’s total tourism. CBT is therefore recognised as the important strategy for employment provision. It not only supports lay people to earn their income for living, it also distributes income to the local areas [14]. Thailand has previously promoted CBT through a variety of projects including One Tambon One Product (OTOP), a project of 12 must-visit secondary cities. OTOP tourism project can generate revenue in the amount of THB 3.5 billion from 2.9 million tourists [15]. This project can distribute income to 19,385 communities. It can be said that CBT is one type of eco-tourism that drives local participation and promotes social equity and redistributive justice [16].

CBT was considered the most important approach for the sustainability development of the post-pandemic tourism sector [6]. The Thailand Tourism Promotion Action Plan for the Year 2021 aimed to create a future of Thai tourism based on safety and a sustainable society [17]. Its action plan focused on increasing the ability to ensure safety and build confidence in strong public health measures, especially for CBT located in rural areas. Nakhon-Si-Thammarat province establishes varied tourist destinations in southern Thailand with several notable CBT and cultural attractions [18]. CBT services include homestays, restaurants, and recreational activities establishments. Those services have to strictly comply with the SHA standard to ensure safety and prevent disease outbreaks in the community. Nevertheless, various features and types of CBT services probably make it indistinct for operators how to adapt and follow the restricted SHA protocol. In addition, the unique attributes of CBT operations may affect the implementation of the SHA, such as public health knowledge of community members, community climate, or leaders. Because SHA is a public health-based measure that has to be implemented by the member of CBT, thus, the community readiness assessment is a necessary method to operate at the starting point of public health intervention [19].

Community readiness is an essential strategy for preventive health programmes due to it could lead to better community participation [19,20]. It can be said that the level of community readiness should always be considered and assessed in implementing health interventions. When necessary, its level may be improved by increasing awareness and engagement prior to implementing the interventions [19,20]. The Community Readiness Model (CRM) is a method to assess community readiness that has been widely used in health promotion and prevention in many communities [19,21,22,23]. Its principle comprehensively considers readiness by exploring current community knowledge efforts, present leadership, community climate, education on the issue, and available resources for the readiness of public health intervention or measures [20]. Many previous studies have explored the knowledge, attitudes, and practices toward COVID-19 prevention and public health measures to prevent and control COVID-19 in community settings [24,25,26,27]. There was no readiness assessment of the public health measures implementation for COVID-19 prevention among the CBT, although their tourism activities and community characteristics affected the health and safety of tourists and community members. This study aimed to explore the readiness of SHA measures for COVID-19 prevention in Nakhon-Si-Thammarat CBT, Southern Thailand using the Community Readiness Assessment. The study would provide explicit information on the readiness of the CBT and could propose explanations to improve the implementation arrangement of the public health measures in the tourism community at both aspects of CBT and relevant organisations. As well, the finding of this study would support effective interventions on public health measures because it requires the cooperation of the community for sustainable practice.

## 2. Materials and Methods

### 2.1. Study Design and Setting

A qualitative study with a cross-sectional community readiness assessment was carried out to determine the level of readiness regarding the SHA implementation as this approach allows us to understand the participants’ views and experiences about study issues in deep detail [19]. The study was conducted on three types of CBT in Nakhon-Si-Thammarat province, southern Thailand: rural mountain, rural fishing, and orchard villages. These three types of establishments included recreational activities, restaurants, and homestays. All CBTs are operated by community enterprises or supported by local authorities.

### 2.2. Data Collection

Fifteen participants were purposively invited to face-to-face semi-structured in-depth interviews. The inclusion criteria were: (1) establishing owner or leader of CBT club; (2) 20 years old and above; (3) at least one year of experience in operating a CBT club, and (4) willingness to participate in this study. Snowball sampling was then employed to further increase the number of key informants. We asked previous key informants to direct us to other key informants who may know more about issues being investigated. This can gain insightful information related to study objectives. Data collection was carried out from February to May 2021.

All participants were informed about the study background, its objectives, and benefits of participation in terms of the SHA perception and improvement method before their in-person interviews. A written consent form was then obtained. The interviews took place where they felt comfortable, which was mostly at the community hall center. Their native languages were used in the communication as it helps us to gain insight into the story and phenomenon being investigated better. However, the conversation lasted for between one and two hours and a digital voice recorder recorded the conversation for data analysis. The interviews and discussions were translated and transcribed into English by the researchers. To analyse and interpret the data, thematic analysis according to the CRM was subsequently employed. According to thematic analysis method, the researchers can take part in the process of interpretation and induction during fieldwork data collection. Additionally, this is consistent with the nature of qualitative studies where theories are developed by emerging information rather than testing prevailing theories and this can lead to trustworthiness and rigor [28].

We initially read through the manuscript to get familiar with the qualitative data and performed coding using colour pen and repeated coding several times. Later we grouped them into sub-theme and themes. Axial coding was carried out on linking between sub-theme and theme developing for meaningful categories. Selective coding was applied to form final theme to develop core stories of study. We stopped data analysis when no new theme emerged. To ensure the triangulation of the study, the member-checking technique was monitored. The transcripts were confirmed by participants for truthfulness. The draft was given to participants to see whether the researchers interpret or understand their views and experience sensibly. Additionally, they can provide some feedback or message that might have been missing. Group review was used to ensure the rigor: the transcript was initially analysed by the first author and then reviewed by the second and third author.

### 2.3. Community Readiness Assessment (CRA)

SHA is the public health measure for adapting tourism services during the COVID-19 pandemic, certifying tourism establishment and services, and ensuring safety and health standards. The basic protocol of public health measures includes health behaviour improvements and sanitation practices in (1) building and equipment hygiene; (2) arrangement of cleaning equipment to prevent the spread of germs; (3) protection activities for tourism operators.

The readiness scoring system of SHA implementation was adopted from the CRM handbook [22]. The main issue of readiness assessment was “The implementation efforts of SHA protocol for COVID-19 prevention in the CBT”. The five dimensions applied to analyse the main themes under the CRA scoring system: (A) community knowledge of the efforts; (B) leadership; (C) community climate; (D) community knowledge about the issue; (E) resources related issues. Opened-end questions were used to determine those dimensions mentioned above such as “How much did the CBT entrepreneurs know about the SHA protocol or it is a public health measures?”, “How do the CBTs’ leaders support SHA implementation and CBT members participation?”, “How does the community support the efforts of SHA and are there any barriers?”, “How much do the CBT members know about SHA protocol and its benefit?”, and “What are the supporting resources for SHA protocol and implementation?”.

The transcription of each main theme was then scored and described according to the CRM handbook [22]. The overall readiness stage was classified using a nine-point rating scale. In doing so, the transcripts were read and categorised into themes and subthemes by stage of readiness from text itself through repeated reading, and then scored by two researchers independently. Raters then discussed and developed a consensual score for each dimension if a mismatch in the score was presented as recommended by the CRM handbook. Later, the comprehensive score was calculated as an arithmetic mean of the five dimensions. The community readiness stage was subsequently assigned according to the mean, and the CRM stage description defined the readiness stage.

## 3. Results

### 3.1. Characteristics of CBT Settings and Participants

The interviews were conducted among 15 key informants from recreational activities, restaurants, and homestays in the study setting (Table 1). A greater percentage of participants was female (53.3%). The mean age was 49.0 ± 13.7 years; some lived in the area for more than 40 years (46.7%); about two-thirds of participants completed higher education (60.0%). The mean business duration of the establishments within the tourist attractions was 7.3 ± 4.5 years.

### 3.2. Community Readiness Stage

The community readiness scores of the three settings ranged from 3.94 to 4.70. Figure 1 demonstrates the difference in CRA scores in five dimensions. The leadership, community knowledge, and resources of the three settings seem similar to the readiness stages. It also showed that leadership was the highest score of readiness (5.3 ± 0.7) in all communities. The overall score for community readiness for the main issue was 4.32 which is in the pre-planning stage (Table 2). This was defined as a clear recognition that the leader must activate some activities of the SHA protocol, and there may be a group that addresses the measures in CBT. However, the implementation efforts of the measures are not specified or provided details. CBT’s climate is beginning to acknowledge the necessity of dealing with the implementation problem that may be occurred while public health protocol was implemented.

### 3.3. Theme One: Community Knowledge of the SHA Implementation Efforts

The average scoring of this dimension was 3.9 (Table 2), which means at least some CBT members have heard of local efforts for the SHA implementation. Some of the leader participants knew that SHA is a tourism measure during the COVID-19 pandemic. They also recognised the benefit of SHA operation for the CBT business. All participants who participated in the study did not understand the SHA protocol. Nevertheless, they stated that there was no detail of SHA protocol implementation in community practice.

#### Subtheme: Inadequate Knowledge of SHA Protocol, Activities, and Details

According to the key informants from the mountain village, they are acquainted with the public health and social measures for COVID-19 prevention. The provincial tourism organisation endorsed the SHA programme to ensure safety and health for the tourists. However, there has been limited knowledge of the SHA implementation efforts in CBTs.

“*The provincial tourism has also come to our club, focusing on the COVID-19 prevention actions, such as temperature checks, using alcohol to clean our hands, sanitizing our areas, and wearing face masks. Actually, we do it in our daily lives and when we receive tourists during COVID-19 pandemic in the early year. I need to do SHA standards, but how can I start? I and our team do not clear for the action process…*”.(Participant 10)

### 3.4. Theme Two: Leadership

The average scoring of the leadership dimension was 5.3 (Table 2), indicating that at least some of the leadership is participating in developing, improving, or implementing the SHA implementation process. The community leaders showed strong motivation and enthusiasm for the SHA implementation. They stated that the team leaders may initiate the starting point preparation, which is a salient factor for participatory and successful implementation. Participants also shared that they had peer groups of village health volunteers (VHV) for tourism management during the COVID-19 pandemic.

#### 3.4.1. Subtheme: Strong Motivation of the Team Leaders

The strong relationship between the leaders and members is an important feature of CBT. The leaders of such CBT groups have close contact with their CBT members so that they can directly respond to the SHA efforts.

“*I try to plan things according to its standards, although I didn’t know the details. I think SHA will make the benefit to our community not only for COVID-19 prevention but also the benefit of tourism income.*”(Participant 1)

“*I think we have to start from ourselves…other members don’t know SHA well too, whether its’ going to be done initially, it must be done for an example first, then the members are able to follow.*”(Participant 10)

#### 3.4.2. Subtheme: Peer Group of Village Health Volunteer

The SHA measures drive by the local government, but the CBT has to implement the programme themselves. During the COVID-19 pandemic, VHV participates in the prevention action with CBT activities. Community health personnel play their role as supporters that could effectively join with public health measures in CBT settings.

“*At the early phase of pandemic, I didn’t know how to prevent the COVID-19 in my homestay services; however, VHV came to join information and prevention activities with our CBT, such as notifying the community epidemic, communicating the public health measures…*”.(Participant 12)

### 3.5. Theme Three: Community Climate

The average scoring of this dimension was 4.1 (Table 2), signifying that some CBT leaders believe that this issue is a concern of their members, and a few may be participating in the developing, improving, or implementing efforts process. There are various types of establishments in the CBT destination. The CBT club could support the SHA efforts, while obstacles to the SHA implementation efforts were the inaccessibility of the small CBT entrepreneurs.

#### 3.5.1. Subtheme: Strong Community Engagement

Most CBT entrepreneurs are established by the community club. Engagement in business operations and participation could drive the improvement process. The team leader has the potential to induce the other members to participate in public health measures.

“*I believed that our club members could drive the SHA efforts. We have to set an example and then expand it to the various activities of the community members because all sectors have to do the same*”(Participant 8)

#### 3.5.2. Subtheme: Inaccessibility of the SHA Efforts

There are different types of CBT establishments in tourism destinations. The small enterprises do not access the SHA measures because the owner has limited knowledge and information about the protocols and registration process.

“*I perceived the SHA protocol, but our members such as small shops and restaurants may not access. I thought the provincial tourism government should communicate more information to the CBT groups. This action might initiate before the full implementation of SHA.*”(Participant 1)

### 3.6. Theme Four: Community Knowledge about the Issues

The average score of community knowledge about SHA implementation was 4.4 (Table 2). It demonstrated that at least some community members are aware that the SHA implementation occurs locally. The CBT leader reflected that there is the SHA implementation in some CBT settings. The operation of the SHA protocol was supported by provincial tourism and public health organisation. Then, SHA-certified practices are approved with evidence of action. The badge of certification represents the safety and sanitation of community tourism management and compliance with the country’s public health measures.

#### 3.6.1. Subtheme: The Support of Local Tourism and Health Organisation

During the initiation phase, the local tourism organisation has to support knowledge, resource, and process implementation of the SHA registration through an online platform. The SHA protocol varies according to the type of establishment. The CBT owner has to learn about the guidelines and suggestions of government officers.

“*I haven’t started the process yet. But I know that other places have already done it, a sample place. There is a tourism operator as a mentor and training from public health officials.*”(Participant 3)

#### 3.6.2. Subtheme: The SHA Badge Represents the Safety and Sanitation of Community Tourism Management

The SHA implementation was started by registration with the evidence of activities. The TAT would approve the successful registration and give the badge that guaranteed the safety and sanitation of community tourism management. The CBT leaders are required to comply with SHA implementation because of its benefit.

“*I used to do homestay standards from travel agencies. SHA as public health standards were met during this pandemic. I would be ready to do it, although its’ protocols are complex.*”(Participant 15)

### 3.7. Theme Five: Resources Related Issues

In relation to resources for the efforts, the score was reported as 3.9 (Table 2), showing that there are some resources that could be used for further efforts. There is little or no action to allocate these resources for implementation to this issue. There were sufficient resources for SHA implementation, such as community funds, academic institutes networks, public health professionals, and local wisdom. However, there were no actions of resource utilisation for the SHA effort. The participants need local tourism and public health personnel to assist with the operation at the initiation phase. They also supposed that peer assessment was an important determinant for implementation as teamwork is vital for success and sustainability.

#### Subtheme: Personnel Support and Strengthen Peer Assessment Team

During the initiation phase of implementation, the CBT leaders needed guidance from tourism organisations or experts. They required the SHA process training, for instance. They expected to join the coaching team for the SHA implementation and development in their local village. The CBT members indicated that peer assessment of the SHA activities would bring about the learning process of development.

“*I think we already have enough resources. In the community, we can support everything—materials, equipment, and staff. However, that would be great, if we have a training programme before starting or experts to assist with the implementation.*”(Participant 10)

### 3.8. Limitations

This study has strength in applying the theoretical models to conceptually identify the readiness to implement the public health measures against COVID-19 from a community tourism perspective. However, it does not conduct in other parts of the country to generate more collective factors due to the earlier phase of the COVID-19 pandemic. Gathering information only from key informants through in-depth interviews might affect the transferability of the result.

## 4. Discussion

The COVID-19 pandemic impacted many sectors, including the business income of tourism. CBT was considered the most salient policy for the sustainability enhancement of the post-pandemic tourism sector, especially in rural areas of Thailand and many developing countries. Thailand set the SHA, public health-based measures to be an important strategy for the tourism sectors amidst the COVID-19 pandemic [29]. The readiness characteristic of CBT is a significant factor in the effectiveness of the SHA programme implementation. Therefore, we applied the CRA to investigate CBT’s readiness to implement the SHA, a public health-based measure in light of the new normal practices amidst the COVID-19 pandemic prevention. The results of this study highlighted that community readiness currently corresponds to the pre-planning stage, which is a clear recognition of the SHA and its benefit among the CBT leaders. There were predominant leadership roles and a community climate, which could support the implementation better. Some related activities originated from the CBT leaders, such as seeking SHA knowledge and experts. However, the efforts are not focused on detailed planning, and only the CBT leaders expressed the challenges for the SHA implementation. Research findings also showed high awareness of SHA among CBT leaders and disclosed sufficient community resources, but the SHA implementation process is complex for operations. There was a need for SHA protocol information, personnel competency development, expert coaching, and training before and during programme implementation.

According to the characteristics of CBT, it is operated by community members in rural areas. The study revealed that they have inadequate knowledge efforts of the SHA protocol and practice among CBT members. Although the CBT leaders have a strong motivation for the benefit of SHA in terms of income benefit and service accreditation, this situation is probably a key barrier to effective implementation [27,30,31], The findings indicated that local health authorities and tourism organisations should prepare for action by organising capacity-building activities such as educational training to develop CBT members who operate in a systematic and two-way communication to make SHA operations more efficient and effective [32].

In addition, the finding of this study illustrates the strong motivation of the CBT team leaders to achieve to SHA standard. Moreover, the village health volunteers coordinated the public health activities, such as tourist screening during the COVID-19 pandemic. This situation reflects the community leadership in public health practice and community human resources that could support the SHA implementation. It also demonstrates strong community engagement of the community leaders which affects CBT’s performance. These results indicate essential features of the CBT model arising from participatory processes and the need for collaborative care for the quality of life of community members from all sectors in the area. These findings correspond with many studies that community members’ reinforcement of community leadership and engagement affected community responsibility [33,34,35]. Therefore, institutional public health campaigns should highlight compassionate attitudes toward SHA benefits in the CBT and the whole community. It can promote public health attitudes and practices through SHA among CBT members. This process may improve effective adherence to public health measures such as SHA [35].

The results of community knowledge indicate the support of local tourism and public health organisations to the CBT leaders at the initiation stage, by providing with the SHA badge for enthusiasm and adherence for SHA operation. Nevertheless, the small CBT did not access SHA knowledge and efforts due to information access. It perhaps affects community awareness and would be the barrier of the implementation readiness. The previous study mentioned that insufficient information regarding the intervention process, mainly among community members, may cause less compliance with community intervention [33]. These findings suggest that public health awareness campaigns should be employed before SHA implementation to improve community awareness about disease prevention and control in the public settings and services. Proper communication about the benefit and morality of the SHA implementation with CBT members may also raise motivation through community members’ knowledge and efforts [36,37].

Consistent with the model arrangement of CBT, they reflect that the resources for SHA implementation, such as finance, equipment, and human resources, are sufficient. SHA implementation requires many tools and equipment. The CBT could seek the material, for instance, advertising media, signs, cleaning equipment disinfectant, and others by the SHA requirement from team members in their community. Likewise, they reveal that community public health personnel, local tourism personnel, and peer are the existing community resources that could support effective implementation. Yet, the CBTs were ready to seek more resources to provide SHA protocol if it is necessary. The existing resources in the local community are recognised as key factors that can upkeep the initiation stage of implementation [37]. This finding differs from other studies, which revealed that financial related resources were the main barriers to accessing public health measures for the prevention of COVID-19 [38]. Still, they requested public health experts, and tourism officers from the government sector to help them with proper SHA arrangements. Those that correspond to such resources could be used for further efforts [20].

In reality, the majority of CBT members encountered difficulties to access the online SHA standard platform. To access this platform, individuals are required to have good computer and digital skills. Most CBT group members who participated in this study are lay people and they are of older generations. The implementation efforts of public health measures should consider socio-cultural characteristics and community contexts [39]. Like with personal change, this problem may bring about a lack of readiness to resolve issues by community leaders and members [22]. Therefore, the tourism agency that manages the implementation of SHA measures must manage communications and practices concerning the social and cultural contexts of the community tourism establishments, including the readiness of the practitioners to make the performance possible, efficient, and sustainable.

## 5. Conclusions

CBTs offered various services and are operated by community members. The implementation of SHA in the CBT industry must be a concern of both CBT competencies and sustainable practices. The finding of this study showed that CBT’s readiness was at the pre-planning stage, which means that there is a clear recognition of the SHA implementation benefit. However, CBT members still have inadequate knowledge of SHA protocol and relevant detail, which is mainly caused by insufficient and inaccessible information. CBTs have sufficient resources, such as materials and teams for SHA implementation. Local tourism organisations should coordinate with the CBT leaders for the quality of tourism arrangements. They still required more public health perspectives to support the SHA implementation, including infectious disease prevention, sanitation management, and health risk communication at the individual and community levels. CBTs require systematic start-up support from accessible channels of SHA information and training before comprehensive implementation. Therefore, our study suggests that local tourism and public health organisations in the local community should provide the appropriate platform for SHA information access, public health-based knowledge of the SHA protocol, and method to manage resources according to the context of CBT services.

The individual, community, and local organisations are required to drive the CBT’s public health and environmental measures. Empowerment and community participation approaches are essential strategies to increase the healthy lifestyle of people. Apart from this, it needs to build collaborations between organisations within the community to provide education and training for people so that they can contribute more to CBT development. This research is highly beneficial to local governments. The local government should have tools to analyse the community’s readiness, feasibility and limitations in implementing public health measures for their tourism community. Moreover, the CBT has unique characteristics which differs from the general assembly; implementing a public health and safety environment must be a concern for sustainable development. To ensure the safety and health of the CBT and their community using the public health-based measures, further study should apply action research to build up the CBT competencies and explore the resilience model, which is more appropriate for various types of CBT services and community readiness.

## Figures and Tables

**Figure 1 ijerph-19-10049-f001:**
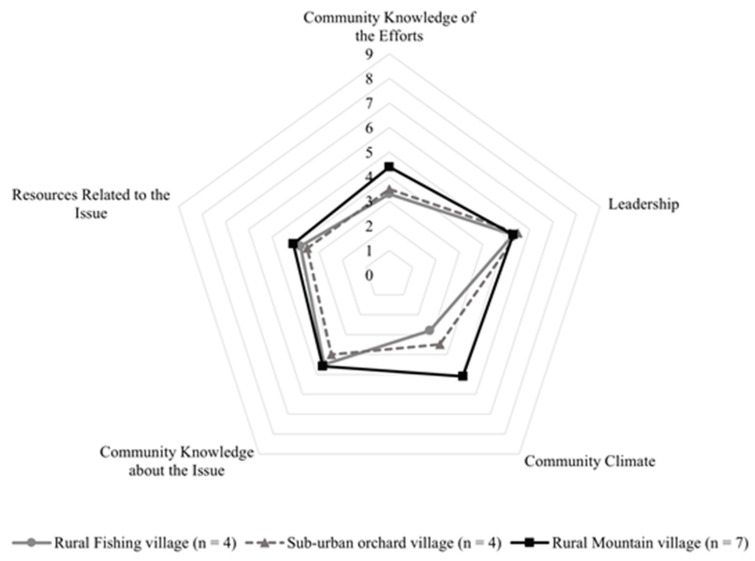
Community readiness score among CBT settings.

**Table 1 ijerph-19-10049-t001:** Characteristics of CBT settings and participants.

	Participant Number	Age	Sex	Establishment Types	Role in CBT	Main/Co-Occupation	Religion
Rural fishing community	1	50	Male	Recreational activity	Head of rural fishing village	Artisanal fisheries	Islam
	2	54	Female	Restaurant	Restaurant owner	Artisanal fisheries	Islam
	3	26	Male	Restaurant	Restaurant owner	Restaurant owner	Buddhism
	4	37	Male	Restaurant	Restaurant owner and Head of rural fishing village	Restaurant owner	Islam
Sub-urban orchard community	5	56	Male	Recreational activity	Souvenir shop owner	Pottery owner	Buddhism
	6	46	Male	Recreational activity	Orchard owner	Orchard owner	Buddhism
	7	46	Female	Recreational activity	Orchard owner	Orchard owner	Buddhism
	8	22	Female	Recreational activity	Head of sub-urban orchard village	Local government officer	Buddhism
Rural mountain community	9	51	Female	Recreational activity	Batik painting shop owner	Restaurant owner	Buddhism
	10	52	Female	Recreational activity	Head of rural mountain village	General	Buddhism
	11	39	Female	Recreational activity	Mushroom farm owner	Mushroom farm owner	Buddhism
	12	69	Female	Homestay	Homestay owner	Homestay owner	Buddhism
	13	68	Male	Homestay	Homestay owner	Homestay owner	Buddhism
	14	64	Female	Homestay	Homestay owner	Homestay owner	Buddhism
	15	55	Female	Homestay	Homestay owner	Homestay owner	Buddhism

**Table 2 ijerph-19-10049-t002:** Community readiness dimension score by main themes and subthemes of the CBTs for safety and health administration implementation.

Main Themes (Community Readiness Dimensions)	Subthemes	CRA Scores (*n* = 15)	
		Mean	Std. Dev.
Community knowledge of the SHA implementation efforts	Inadequate knowledge of SHA protocol, activities, and details	3.9	1.1
Leadership	Strong motivation of the team leaders	5.3	0.7
	Peer group of village health volunteer		
Community climate	Strong community engagement	4.1	1.7
	Inaccessibility of the SHA efforts		
Community knowledge about the issue	The support of local tourism and health organisation	4.4	0.6
	The SHA badge represents the safety and sanitation of community tourism management		
Resources related to the issue	Personnel support and strengthen peer assessment team	3.9	0.6
**Overall readiness level ***		**4.32**	**0.94**
**Stage of readiness**		**Pre-planning**	

* Overall readiness level is calculated by dividing the summation of each dimension mean by five.

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
