# Peer review of "Exploring Readiness towards Effective Implementation of Safety and Health Measures for COVID-19 Prevention in Nakhon-Si-Thammarat Community-Based Tourism of Southern Thailand"

_ijerph, 2022, doi:10.3390/ijerph191610049_

Round 1

Reviewer 1 Report

Thank you for your submission

The article is interesting even if based on a small champion of interviews. For this reason it is worth trying to improve its quality and coherence.

Here the suggestions:

Some specifics and insights will serve to better illustrate the results of this research, which is focused on a limited group of people but can give broader indications for the application and effectiveness of prevention measures in tourism.

The abstract must be revised at the end, based on the observations made for the text as a whole (methodology, recognition of the need for SHA implementation).

Line 56 – please give a more accurate definition of e co-tourism, and an idea of the actual percentage in Thailand, or dimensions of communities living thanks to this tourism, and economic weight of this sector.

Line 82 – this is a promising outcome of your research that you have to better focus – at the end of the article you have conclusions, and you should try to explain how to use and “explanations to improve …. behaviours of the community”

Line 92 – Moreover ….. it requires the community observance …..” . – I don’t think this phrase is correct - the requirement is in the SHA existing regulation itself, the need for which is recognised by the operators themselves. Here the problem is to understand what is really happening in terms of implementation. The impression is that there are rules that are not being implemented, even though the economic resources exist. Perhaps there are partial and uncontrolled applications. Your role as researchers, once you have observed the problems, is to offer some suggestions for improvement in the direction of prevention (in the conclusions of the work).

Line 96 – as I read your paper, the method is qualitative, please explain why you define it mixed-method. Explain also how you did your choice of the three groups and why it is significant for you. à this also linked to study limitations , if your explain better your choice you can find that the conclusion from this study can be applied more widely (may be)

Line 109 – please explain the benefits you mention

Line 118 – please explain “triangulation”

Results – the methodology is applied, but many results are relevant and not  commented or adequately observed. For example you say – line 183 “All participants participated in the study did not understand that the SHA protocol is a public health-based  measure that has to provide for COVID-19 prevention at the community level” – this is a crucial aspect to understand , and connect with the “ enthusiasm for SHA implementation”. This reflect also in the discussion – did they understand what is SHA? – we suppose they did, even if they “did not understand that the SHA protocol is a public health-based

 measure that has to provide for COVID-19 prevention at the community level”

but in

Line 311 – I would say “a clear recognition of the NEED OF SHA implementation

Line 318 – “high awareness and knowledge of SHA” – are you sure?

Line 340-350 – resources are scarce for Covid prevention in genral, but available for SHA, right?

Discussion in general – try to put in order your observations, that are interesting , distinguishing the issues – like training , planning , support , control – trying to provide more practical elements about tools to be produced  , For example “Therefore, the effective public health interventions to improve adherence to social and  public health measures like SHA should couple individual-level strategies in the CBT  leaders and institutional public health campaigns to highlight compassionate attitudes  towards SHA benefits in the CBT and the whole community.” This is a crucial point – how to put it in practice? You can say “As an example, public health campaigns should include …(themes/issues/storis)… ….. …. , the interdisciplinary team to build training and dissemination should include psychologists, anthropologists (or/and others) … should be produced together with community members …. should  include school goers / general practitioners / others …. as testimonials and persons to help disseminating …..

I hope those sUggestion will improve your text and  that you  will proceed with your research, that is useful to promote public health and prevention.

Thank you for your effort and best regards

Author Response

Dear Reviewer,

Re: Exploring Readiness towards Effective Implementation of Safety and Health Measures for COVID-19 Prevention in Nakhon-Si-Thammarat Community-Based Tourism of Southern Thailand: Using a Modified Community Readiness Model

Thank you very much for giving us an opportunity to revise this important piece of workWe have done the revision, taking into account all comments. Our responses are set out via a file attached.

Reviewer 2 Report

1. Why the author choose a mixed-method? Please specify

2. I do not see any qualitative analysis in this study, still, the authors labeled it a mixed method.

3. Justify why you considered a thematic analysis for your qualitative study?

4. In the introduction, please clearly highlight the novelty aspect of this study, in its current form, I am just not convinced how your study advances the existing body of knowledge.

5. Also reconsider your title, it should be precise one.

6. Why Snow ball sampling? What is its relevance?

7. Were interviews audio recorded?

8. What were the steps to transcribe the data?

9. Was Ethical permission obtained to conduct this study?

10. The authors extracted different themes from their analysis, it is suggested to compare and contrast these themes with available literature. It is important to indicate how available literature argues about these themes and their importance.

11. It is also suggest to let the reader know, what were the specific questions that were asked during the interview

12. What was the duration of an interview (you can mention the average time).

Overall summary,

I do believe that this research is relevant and has publication potential after addressing the above points.

Best of luck to the authors 

Reviewer 3 Report

It is necessary to mention in the abstract what this paper brings to the table, especially regarding its academic and practical implications. The paper lacks a critical, creative and innovative approach, making it an essentially descriptive/exploratory work.

The introduction section does not explain in a convincing manner the relevance of the paper, unfortunately. In the introduction the authors should make a better framing of the theme. Therefore, the introduction should include the theme, the motivation for the choice of the problem, the objectives, the methodology to be followed and the work structure. Thus, a contextual discussion of the work must be done, justifying the theoretical, social and practical importance of the research problem.

Furthermore, I strongly believe that there is a lack of information that supports the state of the art., i.e. a theoretical framework that explains the reasons of the research.

It remains unexplained why the Community Readiness Model (CRM) was selected by the authors as the theoretical basis for their study. The study design (based on that theoretical framework) remains unclear. Moreover, in the title of the paper appears “Modified Community Readiness Model”. But, the article does not include the CRM modification!!!!

“A mixed-method approach was adopted for this study” (line 17 and 96! What are these methods?

What are the objectives of the paper? (line 109)

Suggestions for further research must be more developed.

The use of academic English language would need improvement.

The paper were not carried out in accordance with the guidelines for submission of the review (especially references in the text). 

After reading the whole article the motivation for writing paper remains unclear to me. It should be helpful if you can add more information about this aspect.

Round 2

Reviewer 1 Report

Thank you for your revisions, the text is now ready for publication

Reviewer 2 Report

The authors have met my concerns. I recommend the publication of this manuscript. 

Best Regards 

Reviewer 3 Report

The authors made all the changes suggested in the initial revision of the paper in a satisfactory manner, significantly improving the previous version of the paper.

Congratulations to the authors!